# Combined Administration of Escitalopram Oxalate and Nivolumab Exhibits Synergistic Growth-Inhibitory Effects on Liver Cancer Cells through Inducing Apoptosis

**DOI:** 10.3390/ijms241612630

**Published:** 2023-08-10

**Authors:** Vincent Chin-Hung Chen, Shao-Lan Huang, Jing-Yu Huang, Tsai-Ching Hsu, Bor-Show Tzang, Roger S. McIntyre

**Affiliations:** 1Department of Psychiatry, School of Medicine, Chang Gung University, Taoyuan 33302, Taiwan; hjcch@cgmh.org.tw; 2Department of Psychiatry, Chang Gung Medical Foundation, Chiayi Chang Gung Memorial Hospital, Chiayi 61303, Taiwan; tom98877@cgmh.org.tw (S.-L.H.); a2500720@cgmh.org.tw (J.-Y.H.); 3Institute of Medicine, Chung Shan Medical University, Taichung 40201, Taiwan; 4Immunology Center, Chung Shan Medical University, Taichung 40201, Taiwan; 5Department of Clinical Laboratory, Chung Shan Medical University Hospital, Taichung 40201, Taiwan; 6Department of Biochemistry, School of Medicine, Chung Shan Medical University, Taichung 40201, Taiwan; 7Mood Disorders Psychopharmacology Unit, University Health Network, University of Toronto, Toronto, ON M5T2S8, Canada; roger.mcintyre@uhn.ca; 8Department of Psychiatry, University of Toronto, Toronto, ON M5T1R8, Canada

**Keywords:** liver cancer, nivolumab (Niv), escitalopram oxalate (Esc), synergistic effect

## Abstract

Liver cancer is one of the most lethal malignant cancers worldwide. However, the therapeutic options for advanced liver cancers are limited and reveal scant efficacy. The current study investigated the effects of nivolumab (Niv) and escitalopram oxalate (Esc) in combination on proliferation of liver cancer cells both in vitro and in vivo. Significantly decreased viability of HepG2 cells that were treated with Esc or Niv was observed in a dose-dependent manner at 24 h, 48 h, and 72 h. Administration of Esc (50 μM) + Niv (20 μM), Esc (75 μM) + Niv (5 μM), and Esc (75 μM) + Niv (20 μM) over 24 h exhibited synergistic effects, inhibiting the survival of HepG2 cells. Additionally, treatment with Esc (50 μM) + Niv (1 μM), Esc (50 μM) + Niv (20 μM), and Esc (75 μM) + Niv (20 μM) over 48 h exhibited synergistic effects, inhibiting the survival of HepG2 cells. Finally, treatment with Esc (50 μM) + Niv (1 μM), Esc (50 μM) + Niv (20 μM), and Esc (75 μM) + Niv (20 μM) for 72 h exhibited synergistic effects, inhibiting HepG2 survival. Com-pared with controls, HepG2 cells treated with Esc (50 μM) + Niv (20 μM) exhibited significantly increased sub-G1 portion and annexin-V signals. In a xenograft animal study, Niv (6.66 mg/kg) + Esc (2.5 mg/kg) significantly suppressed the growth of xenograft HepG2 tumors in nude mice. This study reports for the first time the synergistic effects of combined administration of Niv and Esc for inhibiting HepG2 cell proliferation, which may provide an alternative option for liver cancer treatment.

## 1. Introduction

As the sixth most common cancer and the third leading cause of cancer-related death, liver cancer is a global health problem with high incidence rates, especially in Asian and African countries [1,2,3]. The treatment of liver cancers encounters many difficulties due to the rapidly proliferative and aggressive nature of the disease [4]. In addition, the pathogenic mechanisms of liver cancers are numerous and complicated, and the strategies for treating live cancers are strictly dependent on the stage of tumor and liver functions [5]. The above-mentioned findings point out the predicament of liver cancer in tailored medical therapy [5]. Accordingly, the overall prognosis of liver cancer is poor, with a 5-year survival rate less than 20% [6]. Although many methods have been adopted for liver cancer treatment, such as surgery, transcatheter arterial embolization (TAE), radiotherapy, chemotherapy, and targeted medicines, the efficacy of these treatments remains very limited in advanced liver cancer [7].

In recent decades, cancer immunotherapy has become recognized as a highly anticipated strategy for cancer therapy [8,9]. The investigation of therapeutic approaches such as adoptive cell immunotherapy, for instance, chimeric antigen receptor T cells and T cell receptor engineered T cells for liver cancers, are developing rapidly [10]. In addition, many investigators have dedicated their research to developing immune checkpoint inhibitors that can relieve immune checkpoint blockage and restore immune activity against tumors [8,9]. Various approved immune checkpoint inhibitors, such as ipilimumab against T lymphocyte CTLA-4 and nivolumab/pembrolizumab against T lymphocyte PD-1, have been widely used in treatment for liver cancer [11,12,13]. However, inhibition of immune checkpoints during ipilimumab, nivolumab, or pembrolizumab treatment often causes adverse events such as hepatitis, dermatitis, pancreatitis, pneumonitis, enteritis, and arthritis [14]. In liver cancer patients receiving immunotherapy, autoimmune disorders such as type 1 diabetes mellitus, myasthenia gravis, and rheumatoid arthritis have been reported [14].

Escitalopram oxalate (Lexapro^®^) is a superior SSRI used to treat major depressive disorder (MDD) and anxiety, and has revealed favorable tolerability and less adverse symptoms in various randomized and double-blind controlled studies [15,16]. Recently, mounting evidence has indicated the anticancer potentials of escitalopram oxalate [17,18,19,20,21]. Indeed, escitalopram oxalate has been demonstrated to inhibit the proliferation of GBM cells [17,18] and non-small cell lung cancer (NSCLC) cells [19] by inducing apoptosis or autophagy. Similar results were reported in a nationwide population-based study, indicating that administration of escitalopram oxalate is associated with decreased HCC risk [20]. Moreover, a very recent study involving both bench work and a nationwide population-based cohort study reported that escitalopram oxalate inhibits the proliferation of liver cancer cells by inducing autophagy and that its use involves a reduced HCC risk [21].

Chemotherapy and immunotherapy in anticancer treatment may cause various adverse events. such as drug resistance and toxicity, leading to a negative impact on therapeutic efficacy [22]. Notably, a systematic review and pooled analysis have both noted that combinational treatments of immunotherapy and chemotherapy exhibit encouraging anti-tumor potential and an acceptable toxicity profile in patients with advanced non-small-cell lung cancer [22]. Therefore, combinational use of multiple anticancer drugs may improve treatment efficacy by inducing synergistic effects and overcoming adverse influences such as drug resistance and toxicity [22]. Because escitalopram oxalate has been reported to have potential against several cancers, the current study adopted combinational use of low-dose escitalopram oxalate and nivolumab to verify the resulting synergistic inhibitory potential against liver cancer both in vitro and in vivo.

## 2. Results

### 2.1. Effects of Escitalopram Oxalate and Nivolumab on Survival of HepG2 Cells

To investigate the effects of escitalopram oxalate and nivolumab on liver cancer cells, HepG2 cells were treated with different concentrations of escitalopram oxalate and nivolumab and the cell viability was detected (Figure 1). Significantly decreased survival of HepG2 cells treated with escitalopram oxalate was observed in a dose-dependent manner, with IC_50_ values of 137.6 at 24 h, 82.1 at 48 h, and 78.5 μM at 72 h (Figure 1A). Similar results were observed for nivolumab, which significantly inhibited the viability of HepG2 cells in a dose-dependent manner; the IC_50_ values were 13.6 at 24 h, 16.3 at 48 h, and 21.8 μM at 72 h (Figure 1A).

### 2.2. Combinational Use of Escitalopram Oxalate and Nivolumab Synergistically Decreases the Viability of HepG2 Cells

HepG2 cells were treated with low-dose escitalopram oxalate (50 μM or 75 μM) and different doses of nivolumab to verify the synergistic effects of combinational use of escitalopram oxalate and nivolumab on inhibiting the survival of HepG2 cells. Significantly decreased viability of HepG2 cells was observed in combinational use of escitalopram oxalate (50 μM or 75 μM) with different concentrations of nivolumab at 24 h, 48 h, and 72 h (Figure 1B). Furthermore, CompuSyn (CompuSyn Inc., Paramus, NJ, USA) was used to calculate the drug pairing’s combination index (CI) values and determine the pharmacological interaction between escitalopram oxalate and nivolumab (Figure 2). As shown in Figure 2, the CI values for combinations 1 to 8 were 1.01687, 1.15940, 1.18695, 0.76961, 1.25610, 0.92150, 1.16051 and 0.83844 at 24 h (Figure 2A), 0.95880, 1.29805, 1.06676, 0.54824, 1.13257, 1.24239, 1.08036, and 0.50689, respectively, at 48 h (Figure 2B) and 0.89171, 1.09178, 1.11474, 0.67150, 1.00425, 1.06352, 1.07250, and 0.63072, respectively, at 72 h (Figure 2C). The following combinations of escitalopram oxalate (Esc) and nivolumab (Niv) showed antagonism (CI > 1) at 24 h: 50 μM Esc + 1 μM Niv, 50 μM Esc + 5 μM Niv, 50 μM Esc + 10 μM Niv, 75 μM Esc + 1 μM Niv, and 75 μM Esc + 10 μM Niv. Additionally, antagonism was shown for the combinations of 50 μM Esc + 5 μM Niv, 50 μM Esc + 10 μM Niv, 75 μM Esc + 1 μM Niv, 75 μM Esc + 5 μM Niv, and 75 μM Esc + 10 μM Niv at both 48 h and 72 h. Notably, the administration of 50 μM escitalopram oxalate + 20 μM nivolumab and 75 μM escitalopram oxalate + nivolumab (5 and 20 μM) for 24 h exhibited synergistic effects (CI < 1), inhibiting HepG2 cell viability (Figure 2A). Similar results were observed for 50 μM escitalopram oxalate + nivolumab (1 and 20 μM) and 75 μM escitalopram oxalate + 20 μM nivolumab at 48 h, exhibiting synergistic inhibitory effects (CI < 1) on the viability of HepG2 cells (Figure 2B). At 72 h, 50 μM escitalopram oxalate + nivolumab (1 and 20 μM) and 75 μM escitalopram oxalate + 20 μM nivolumab exhibited synergistic effects (CI < 1), inhibiting HepG2 survival (Figure 2C).

### 2.3. Combinational Use of Escitalopram Oxalate and Nivolumab Significantly Induces Apoptosis in HepG2 Cells

To measure the portion of different cell cycle stages of HepG2 cells in the presence of 50 μM escitalopram oxalate and 20 μM nivolumab, flow cytometry analysis was performed (Figure 3). A significantly increased sub-G1 portion (26.6%) was observed in HepG2 cells treated with 50 μM escitalopram oxalate and 20 μM nivolumab in combination compared to control as well as to those treated with 50 μM escitalopram oxalate (6.2%) or 20 μM nivolumab (8.2%) alone (Figure 3A). The quantified results of the sub-G1 portions are shown in Figure 3B. Additionally, annexin V/PI double staining analysis was adopted to confirm the involvement of apoptosis in HepG2 cells treated with 50 μM escitalopram oxalate and 20 μM nivolumab in combination (Figure 4). Compared with the control cells and cells treated with escitalopram oxalate (14.2%) or nivolumab alone (19.8%), the HepG2 cells treated with 50 μM escitalopram oxalate and 20 μM nivolumab in combination exhibited significant apoptosis (41.1%) (Figure 4A). The quantification results are presented in Figure 4B. Significantly increased cleaved poly-(ADP-ribose) polymerase (PARP) was detected in the HepG2 cells treated with 50 μM escitalopram oxalate and 20 μM nivolumab (Figure 4B).

### 2.4. Combinational Use of Escitalopram Oxalate and Nivolumab Inhibits the Proliferation of Xenograft HepG2 Tumors in Nude Mice

To investigate the effects of combinational use of escitalopram oxalate and nivolumab in vivo, xenograft HepG2 cell tumors were generated in BALB/c nude mice. Significantly decreased xenograft tumor volumes were detected in mice treated with escitalopram oxalate (2.5 mg/kg) and nivolumab (6.66 mg/kg) in combination compared to the control group and to mice treated with 2.5 mg/kg escitalopram oxalate or 6.66 mg/kg nivolumab alone (Figure 5A,B). Accordingly, apparently decreased PCNA (brown signal) and increased TUNEL positive signals (brown signal) were observed in the sections of xenograft tumors from mice treated with 2.5 mg/kg escitalopram oxalate and 6.66 mg/kg nivolumab in combination compared with mice treated with 2.5 mg/kg escitalopram oxalate or 6.66 mg/kg nivolumab alone (Figure 5C).

## 3. Discussion

Nivolumab (Opdivo^®^), known as an anticancer drug belonging to immune checkpoint inhibitor (ICI), is a fully human immunoglobulin G4 monoclonal anti-body against programmed cell death receptor 1 (PD-1) [23]. The action mechanism of nivolumab is to disrupt the signaling of the PD-1 immune checkpoint and thereby recover the anticancer activity of suppressed effector T cells [23]. In recent decades, evidence has indicated that nivolumab can be used widely for treatment of various cancers, including liver cancer [23,24]. Nivolumab can cause T-cell activation by blocking immune checkpoints and elect autoimmune toxicities, which leads to tumor destruction [25]. However, evidence has indicated that single ICI does not meet the expected goals [26,27]. Indeed, worse treatment-associated side effects or common grade 3 adverse events such as pal-mar-plantar erythrodysaesthesia, elevated aspartate aminotransferase, and hypertension were revealed in a randomized multicenter open-label, phase 3 clinical trial of patients with advanced hepatocellular carcinoma [27]. In addition, nivolumab has shown increased risk of various immune-related adverse events such as maculopapular rash, pneumonitis, hepatitis, and infusion-related reactions, suggesting a higher risk when increasing the dosage of nivolumab [28,29]. Immune checkpoint inhibitor (ICI) therapy has emerged as a rising star in cancer treatment. However, despite the success of ICIs, the issue of resistance to ICIs restricts the patients able to achieve durable responses, which has been a major concern [30]. Recent clinical trials such as KEYNOTE-240 and CheckMate 459 in HCC treatment did not achieve their predefined goals for overall survival [31]. Additionally, resistance to ICI treatment, including nivolumab, has been reported in cancers such as ovarian cancer, renal cell carcinoma, and esophageal squamous cell carcinoma [32,33,34]. Therefore, the combination of ICIs with other therapies, such as tyrosine kinase inhibitors (TKIs), chemotherapy, or local therapy, is commonly considered and adopted in cancer therapy [31]. Accordingly, the current study reports that combinational use of nivolumab and escitalopram oxalate exhibits a synergistic effect on inhibiting the proliferation of HepG2 cells in vitro and in a xenograft HepG2 tumor animal model. These findings may provide an alternative option for liver cancer treatment.

Escitalopram oxalate is an anti-depressive reagent exhibiting favorable tolerability and less adverse clinical events [15,35]. In recent decades, administration of escitalopram oxalate has been found to have a negative correlation with various kinds of cancers, such as bladder cancer, renal cancer, and liver cancer [36,37,38]. Similar results have been reported in bench work, noting that escitalopram oxalate inhibits the proliferation of non-small-cell lung cancer cells and brain cancer cells [17,19]. A very recent study indicated the protective effect of escitalopram oxalate on liver cancer via induction of autophagy [21]. In addition, no adverse event was reported by daily administration of 600 mg escitalopram oxalate [17]. Another study reported that an antibiotic regimen containing escitalopram oxalate reveals synergistic inhibitory effects on growth of multidrug-resistant (MDR) bacteria [39]. Indeed, these findings may provide reasonable suggestions that escitalopram oxalate is a favorable option for inclusion in a regimen for HCC treatment. However, further investigations and clinical trials are merited to verify the curative effects of combinational use of nivolumab and escitalopram oxalate on treatment of liver cancer.

Although various SSRIs such as escitalopram oxalate have been reported to exhibit anticancer properties, the potential molecular mechanisms underlying the antitumor activities of escitalopram oxalate are unclear. Notably, a recent review article reported that certain psychiatric drugs, practically valproic acid, fluoxetine, escitalopram oxalate, and the atypical psychiatric drug aripiprazole show anticancer activity against glioblastoma multiforme (GBM), indicating potential underlying molecular mechanisms [18]. Possible mechanisms or signaling pathways involved in the anti-GBM activity of these antipsychotic drugs include voltage-gated ion channels and transporters, src oncogenic tyrosine kinase signaling, epigenetic modification of histone deacetylase, wnt/β-catenin signaling, and autophagy [18]. Interestingly, a pathways analysis study in patients with major depressive disorder (MDD) using the Kyoto Encyclopedia of Genes and Genomes (KEGG) database indicated that wnt signaling, cancer, endocytosis, axon guidance, and MAPK signaling are involved in the mode of action of escitalopram [40]. This finding provides evidence that escitalopram oxalate can influence wnt signaling, which may be involve in the anticancer activity of escitalopram oxalate. Additionally, escitalopram oxalate is known to cause cancer cell death by inducing autophagy [17,21]. Known as type II programmed cell death, autophagy exerts a dual role in tumor-suppressive and tumor-promoting actions in different contexts and stages of cancer development [41]. Moreover, various studies have focused on the development of autophagy as a novel strategy in anticancer therapy [41]. Likewise, escitalopram oxalate has been demonstrated to inhibit the proliferation of various cancer cells, including GBM8401, HepG2, and Huh-7 cells, by inducing robust autophagy [17,21,42]. Altogether, these findings provide rational inferences for explaining the possible mechanisms underlying the anticancer activities of escitalopram oxalate. However, more investigations are required to verify the authenticity and precise processes of the mechanisms involved in escitalopram oxalate-mediated anticancer activities, particularly with respect to liver cancer.

Notably, depression and anxiety are common in liver cancer patients, with a frequency about 25% and 20%, respectively [43,44]. These psychological illnesses and related symptoms have been reported to be associated with worse survival outcomes and quality of life for liver cancer patients and their family [45,46]. Therefore, antidepressants, particularly escitalopram oxalate, are often used in liver cancer patients with depression comorbidity. Although escitalopram oxalate has been considered as a one of the superior SSRIs, with favorable tolerability and less side effects in treatment of patients with depressive and anxiety disorders [15,16], a few adverse events of escitalopram oxalate should be noted, especially in the context of administration of escitalopram oxalate or escitalopram oxalate-containing regimens in cancer patients [47,48,49]. A previous case report indicated that a prostate adenocarcinoma patient receiving a major opiate analgesic exhibited serious serotoninergic symptoms such as diaphoresis, night sweating, tremors, diarrhea, and mydriasis after treatment with a small dose of escitalopram (5 mg/day) [47]. Notably, complete resolution of most serotonin syndromes was observed within 2 days after discontinuation of escitalopram oxalate [47], suggesting a risk of serotonin syndrome with escitalopram oxalate use. Recently, a significantly higher QTc-interval, which is known as a risk marker of dangerous arrhythmias, was reported in breast cancer patients with concomitant use of tamoxifen and SSRIs, especially in the cases of paroxetine, escitalopram, and citalopram, indicating an adverse drug–drug interaction [48]. Indeed, evidence indicates that drug interactions between antidepressants and oncological drugs may lead to lower efficacy of treatment and increased side effects in cancer patients [49]. Hence, drug interactions in concomitant psycho-oncology medications should be approached cautiously to ensure better safety and effectiveness, which definitively requires further investigations to verify the pharmacokinetic properties of the most widely used antidepressants and oncological drugs in order to avoid possible adverse drug–drug interaction events. Although this study reported a synergistic effect of combinational use of escitalopram oxalate and nivolumab in liver cancer, the findings mentioned above necessitate an awareness on the part of clinicians concerning the potential adverse effects of drug interaction between antidepressants and oncological medicines in cancer patients.

The composition of the tumor microenvironment (TME) in HCC is a complex that involves a variety of immune cells, such as immunosuppressive cells, regulatory T cells, tumor-associated macrophages, natural killer cells, cytotoxic CD4 positive T cells, CD8 positive cells, and myeloid-derived suppressor cells [50]. Therefore, animal models that can better recapitulate the TME of HCC are necessary in order to verify the effects and mechanisms of potential regimens against HCC. Considering the interpatient heterogeneity and underlying mechanisms in HCC, modeling the TME of HCC in animals is a daunting challenge [51]. Recently, syngeneic, genetically engineered, and humanized mouse models have been widely used to investigate HCC [52]. Syngeneic HCC mouse models involve allografting HCC cell lines or mouse tumor tissue into an immunocompetent mouse. This model has been suggested as a crucial tool for preclinical tests of immune checkpoint blockade and investigations of the underlying immune mechanism for anticancer medicines in HCC [53,54]. Genetically engineered mouse models (GEMM) are generated by activating oncogenes or silencing tumor suppressor genes, which can provide further insight into the roles of specific genes in HCC [55]. Notably, humanized mouse models are created by implanting patient-derived primary tumors into immunocompromised mice, which is considered a more accurately approach for mimicking the human HCC TME [53,56]. These model have the same genetic heterogeneity, architecture, and local TME as humans, including the tumor-associated stroma and tumor-infiltrated immune cells [56]. Through the use of the HCC mouse models mentioned above, it is possible to further elucidate the precise mechanisms of combinational use of escitalopram oxalate and nivolumab on the TME in HCC, especially the effect of this regimen on tumor immunity.

The therapeutic options for liver cancers, including liver transplant, surgical resection, embolization, stereotactic body radiation therapy, ablation, chemotherapy, and immunotherapy, are mainly based on the overall assessment of the patient’s liver function and tumor status [57]. Over the decades, these treatments of liver cancers have evolved due to progress in medical research and therapeutic technology that have significantly improved the survival rate and quality of life of liver cancer patients [58]. However, there remains an urgent need for methods to treat advanced liver cancer. Notably, a combination of immunotherapy drugs has become a trend in the treatment of advanced liver cancer [59]. However, very limited efficacy of immune checkpoint inhibitors (ICIs) has been reported in nearly 70% of advanced HCC patients [60,61]. This is an issue that needs urgent attention. A recent study indicated that combination strategies exhibit a promising approach for HCC patients who have no response to a single ICI agent or no predictive ICI biomarkers [61]. Similar findings were reported in the MOUSEION-03 study, specifically, that the administration of ICIs may provide a higher chance of achieving complete remission (CR) in cancer patients [62]. Another study reported that the combinational use of various biomarkers of ICIs showed better potential for advanced cancer therapy [60]. These findings suggest that the discovery of novel biomarkers related to ICIs, such as in the gut microbiome, are urgently needed for advanced liver cancer treatments [60].

Several outstanding concerns in this study needed to be mentioned. Although the current study demonstrated that combined administration of escitalopram oxalate and nivolumab inhibits the proliferation of HepG2 cells by inducing apoptosis, no anti-apoptotic molecules such as Bcl-2 (B-cell lymphoma 2) were detected. Hence, further investigation of anti-apoptotic pathways is merited to verify the precise mechanism of combined use of escitalopram oxalate and nivolumab for inhibiting liver cancer cells [63]. Another concern is the antagonistic effects of certain combinations of escitalopram oxalate and nivolumab (Figure 2). The combined use of escitalopram oxalate and nivolumab at different dose combinations exhibited either synergistic or antagonistic effects. In fact, potential drug interactions, especially antagonism, have received intense attention from researchers. An increasing body of evidence indicates the antagonistic effects of various drug-combinations on treatment of cancers such as esophageal cancer and non-small-cell lung cancer [64,65]. Therefore, healthcare professionals should be aware of the doses of concomitant drugs used and request that patients provide all their medications, including prescriptions, over-the-counter medications, and herbal supplements in order to avoid possible drug interactions.

Finally, a few limitations of this study need to be emphasized. First, only a one-dose combination of each drug was conducted in the animal experiments in this study. Combinations of different drug doses should be considered in future experiments; this could provide more options or possibilities in dose adjustment for liver cancer therapy. Second, the current study used a subcutaneous xenograft tumor mouse model, which cannot accurately simulate the human tumor microenvironment. Therefore, better orthotopic liver cancer animal models, as described elsewhere [54,55,56,66], should be adopted in order to verify the effects of the combinational use of nivolumab and escitalopram oxalate.

## 4. Materials and Methods

### 4.1. Cells and Chemicals

All chemical reagents in this study were purchased from Sigma-Aldrich (St. Louis, MO, USA). Escitalopram oxalate and nivolumab were provided by Chiayi Chang Gung Memorial Hospital, Taiwan. The HepG2 cells, a human hepatocellular carcinoma cell line, were obtained from the Bioresource Collection and Research Center (BCRC) and maintained in Dulbecco’s Modified Eagle Medium (DMEM) (Gibco, Brooklyn, NY, USA) with 10% Fetal Bovine Serum (Gibco, Thermo Fisher Scientific, Waltham, MA, USA) in a humidified incubator at 37 °C in a 5% CO_2_ chamber (Thermo Fisher Scientific, Waltham, MA, USA).

### 4.2. Detection of Cell Viability

The survival of cells was measured according to the method based on the reagent of XTT (Biological Industries, Haemek, Israel), namely, Tetrazolium salt XTT (sodium 3′-[1-[(phenylamino)-carbony]-3,4-tetrazolium]-bis(4-methoxy-6-nitro) benzene sulfonic acid hydrate). A total of 3 × 10^3^ HepG2 cells/well was cultured in a 96-well cell culture plate at 37 °C for overnight. After treatment with escitalopram oxalate and nivolumab alone or in combination, 50 μL XTT was added and reacted for 4 h. The optical density of solution was measured with a microplate reader (EnSpire Series Multilabel Plate Readers, PerkinElmer Inc., Waltham, MA, USA).

### 4.3. Calculation of Combination Index (CI)

Combination index (CI) and fraction affected (Fa) were calculated using CompuSyn (CompuSyn Inc., Paramus, NJ, USA) according to a previous publication [67]. The values of CI < 1, =1, and >1 indicate synergism, additive effect, and antagonism, respectively. Values of fraction affected (Fa) smaller than 0.5, indicating lower growth inhibition, were considered irrelevant. Conversely, values of Fa > 0.5 indicated a significant effect.

### 4.4. Flow Cytometry

The sub-G1 portion of HepG2 cells treated with escitalopram oxalate and nivolumab alone or in combination was measured using flow cytometry analysis. First, the cells treated with escitalopram oxalate and nivolumab alone or in combination were harvested by centrifugation and immersed in 70% ethanol. Next, 10 μL propidium iodide (PI) solution was added and reacted for 30 min at dark. After filtering through a 35-μm nylon screen (Stellar Scientific, Baltimore, MA, USA), the cells were analyzed using an FACSCanto II flow cytometer (BD Biosciences, San Jose, CA, USA).

### 4.5. Annexin V Assay

Cell apoptosis was detected by annexin V assay. HepG2 cells were incubated with escitalopram oxalate (50 μM) and nivolumab (20 μM) alone or in combination. The cells were then re-suspended in 100 μL annexin-binding solution containing 5 μL annexin V–fluorescein isothiocyanate and 1 μL of PI. Subsequently, the stained cells were analyzed with a flow cytometry apparatus (FACSCanto II system, BD Biosciences, San Jose, CA, USA).

### 4.6. Immunoblotting

Immunoblotting was used to measure the amount of poly-ADP-ribose polymerase (PARP). Briefly, the cell lysate was collected with a cell scraper blade and centrifuged at 5000 rpm for 15 min at 4 °C. Then, the cell lysate was suspended in lysis buffer (PRO-PREP™, iNtRON Biotechnology Inc., Gyeonggi-do, Republic of Korea) and the protein concentration was determined with a BCA protein assay kit accordingly to the manufacturer’s instructions (Sigma-Aldrich, St. Louis, MO, USA). Subsequently, 20 μg of the quantified protein was electrically separated onto a 10% SDS-PAGE gel. After transferring the proteins onto a PVDF membrane (1000 Alfred Nobel Dr Hercules, California, CA, USA) and blocking the membrane in 5% non-fat milk for 6 h, antibodies against PARP (Proteintech, Rosemont, IL, USA) and β-actin (Sigma-Aldrich, St. Louis, MO, USA) were added and incubated at 4 °C overnight. Then, the membrane was incubated with horseradish peroxidase (HRP)-conjugated secondary antibodies for another hour. After washing the PVDF membrane three times with PBS-tween, the detection of immune complexes was performed using a chemiluminescent substrate kit (EMD Millipore, Burlington, MA, USA) and a chemiluminescence imaging apparatus (GE Healthcare Life Sciences, Pittsburgh, PA, USA). For quantification of the amount of proteins on the blots, Multi-Gauge Software V3.2 (Fujifilm Corp., Tokyo, Japan) was adopted to quantify the image intensity.

### 4.7. Xenograft Tumor

Tween five-week-old male athymic nude mice (BALB/c nude mice) were purchased from the National Center for Experimental Animals, Taiwan, and kept in a specific pathogen-free (SPF) environment with constant temperature and humidity conditions and a half-day light–dark cycle. This study was officially recognized by Institutional Animal Care and Use Committee (IACUC) of Chiayi Chang Gung Memorial Hospital, Taiwan (IACUC approval number: 2020092102). Briefly, a total of 1 × 10^6^ HepG2 cells in 0.1 mL PBS was subcutaneously injected into the flank of nude mice. The mice were randomly separated into four groups, which were respectively administrated PBS, 2.5 mg/kg escitalopram oxalate, 6.66 mg/kg nivolumab, and escitalopram oxalate (2.5 mg/kg)/nivolumab (6.66 mg/kg) by intraperitoneal injections 5 days a week for 4 weeks. The volumes of the xenograft tumors were measured every week with a caliper and the animals were sacrificed after 5 weeks of treatment. The doses of nivolumab and escitalopram oxalate were based on those in previous studies [21,68].

### 4.8. Detection of PCNA and Apoptotic DNA Fragmentation

An automated stainer (Bond Max, Leica, Buffalo Grove, IL, USA) and a terminal deoxynucleotidyl transferase dUTP nick-end labeling (TUNEL) kit (Abcam, Cambridge, UK) were used to detect PCNA expression level and apoptotic DNA fragmentation, respectively. Briefly, the xenograft tumors from different groups of mice were cut into sections of 3 μm thickness. The sections were then processed via dewaxing, rehydration, and proteinase K incubation. The presence of PCNA and TUNEL-positive signals was then detected with commercial kits in accordance with the manufacturer’s instructions. The section images were scanned at 40× magnification with a PANNORAMIC digital slide scanner (3DHISTECH Ltd., Budapest, Hungary).

### 4.9. Statistical Analysis

SAS JMP 7.0 software (JMP, Cary, NC, USA) followed by one-way analysis of variance (one-way ANOVA) and Tukey’s multiple-comparisons test was adopted to perform statistical analysis. The data were shown as mean ± SEM and a *p* value less than 0.05 was considered statistically significant.

## 5. Conclusions

Overall, the current study demonstrated a novel regimen composed of low-dose escitalopram oxalate and nivolumab, exhibiting a synergistic inhibitory effect on proliferation of HepG2 liver cancer cells, in particular the combination of Esc (50 μM) + Niv (20 μM) in vitro and Esc (2.5 mg/kg) + Niv (6.66 mg/kg) in vivo. Esc (50 μM) + Niv (20 μM) significantly induced apoptosis of HepG2 cells, and the combination of Esc (2.5 mg/kg) + Niv (6.66 mg/kg) significantly suppressed the growth of xenograft HepG2 tumors in nude mice. For the first time, this study indicates the potential of combinational use of low-dose escitalopram oxalate and nivolumab for inhibiting liver cancer cell proliferation, which may provide an alternative treatment for liver cancers.

## Figures and Tables

**Figure 1 ijms-24-12630-f001:**
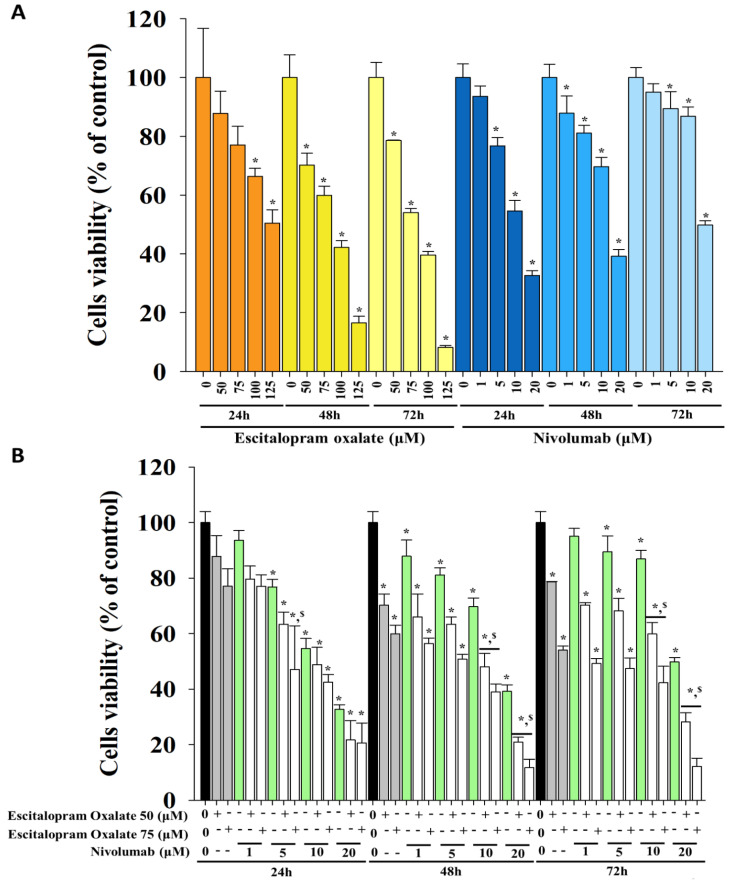
Effects of escitalopram oxalate and nivolumab on the survival of HepG2 cells. (**A**) The viability of HepG2 cells in the presence of escitalopram oxalate and nivolumab at 24 h, 48 h, and 72 h. (**B**) The viability of HepG2 cells in the presence of a combination of escitalopram oxalate and nivolumab at 24 h, 48 h, and 72 h. The symbols * and $ mean statistical significance as compared with control (0 μM) and escitalopram oxalate, respectively. Experiments were performed in triplicate.

**Figure 2 ijms-24-12630-f002:**
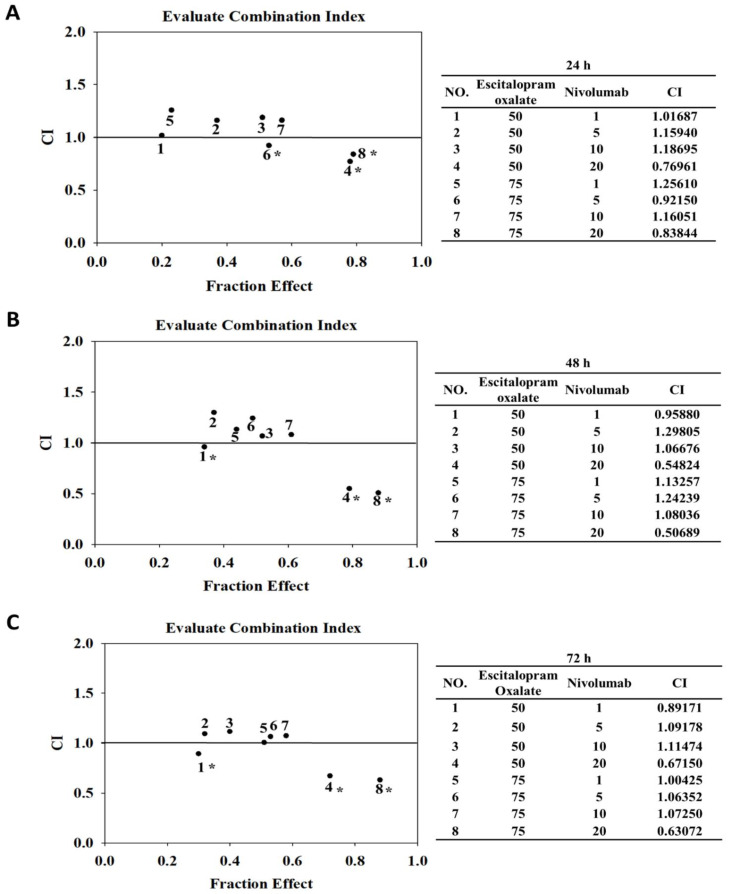
Assessment of combination index (CI). The CI values of combinational use of escitalopram oxalate and nivolumab on the survival of HepG2 cells for (**A**) 24 h, (**B**) 48 h, and (**C**) 72 h. The symbol * indicates synergistic effects. Drug combinations 1–8 are shown in the tables.

**Figure 3 ijms-24-12630-f003:**
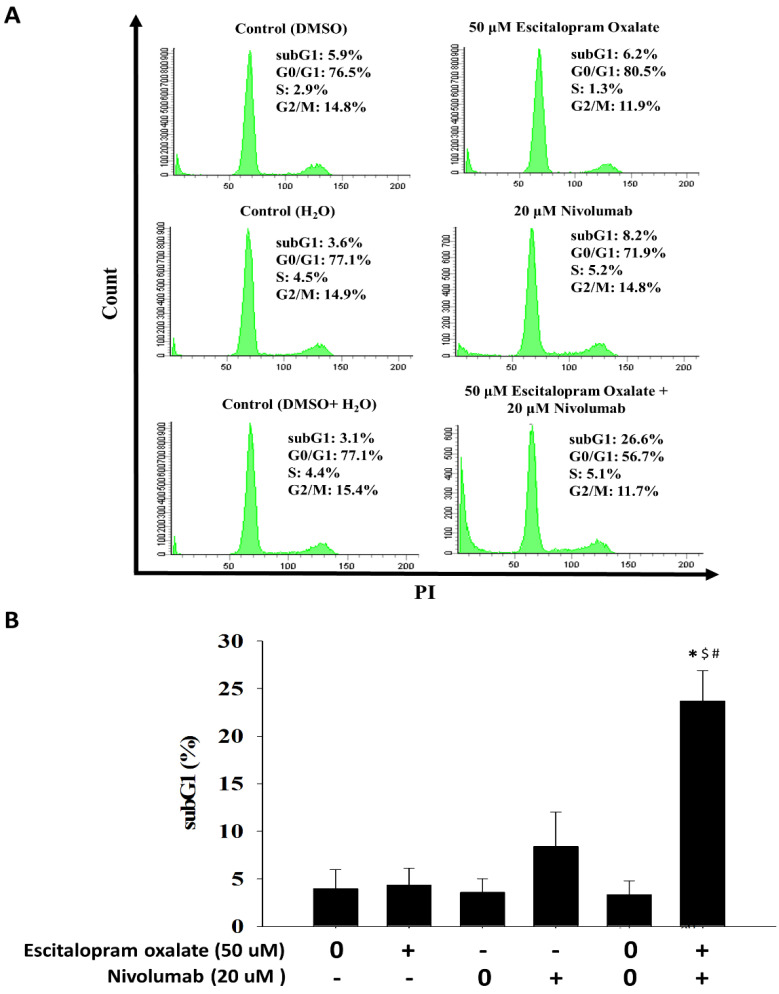
Effects of the combinational use of escitalopram oxalate and nivolumab on the sub-G1 portion and cell cycle in HepG2 cells. (**A**) Representative images of the cell cycle and sub-G1 portion of HepG2 cells treated with escitalopram oxalate (50 μM) and nivolumab (20 μM) alone or in combination for 72 h. (**B**) Quantified results of the sub-G1 portion in HepG2 cells. Bars indicate mean ± SD from three repeated experiments. The symbols *, $, and # mean statistical significance as compared with control (0 μM), escitalopram oxalate (50 μM), and nivolumab (20 μM), respectively.

**Figure 4 ijms-24-12630-f004:**
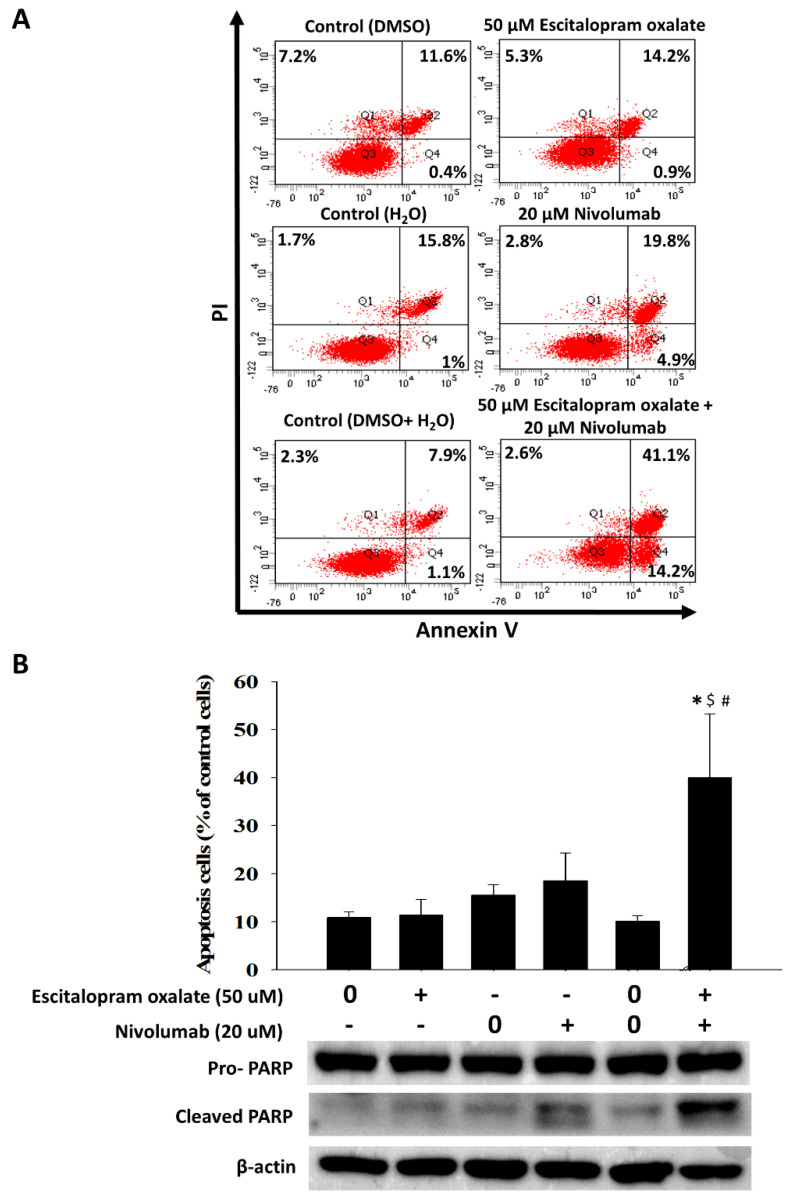
Effects of the combinational use of escitalopram oxalate and nivolumab on apoptosis in HepG2 cells. (**A**) Representative images of annexin V/PI staining and (**B**) percentage of apoptotic cells in HepG2 cells treated with escitalopram oxalate (50 μM) and nivolumab (20 μM) alone or in combination for 72 h. The lower panel shows the expression of cleaved PARP proteins. Bars indicate mean ± SD from three repeated experiments. The symbols *, $, and # mean statistical significance as compared with control (0 μM), escitalopram oxalate (50 μM) and nivolumab (20 μM), respectively.

**Figure 5 ijms-24-12630-f005:**
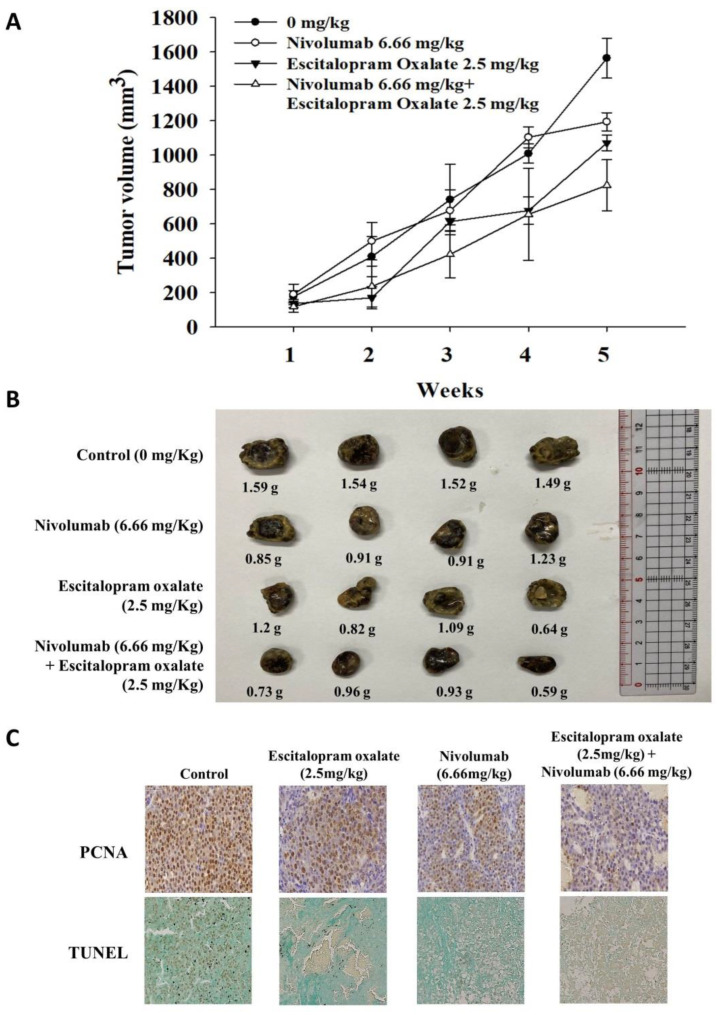
Effects of combinational use of escitalopram oxalate and nivolumab on growth of xenograft tumors in nude mice. (**A**) Volume of xenograft HepG2 tumors in nude mice over a time course pattern. (**B**) Representative images of xenograft tumors excised from mice at the endpoint of the experiment. (**C**) PCNA expression and TUNEL staining of xenograft tumors from different groups of mice. The images of sections are shown at 40× magnification.

## Data Availability

The data presented in this study are available on request from the corresponding author.

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
