# Peer review of "Combined Administration of Escitalopram Oxalate and Nivolumab Exhibits Synergistic Growth-Inhibitory Effects on Liver Cancer Cells through Inducing Apoptosis"

_ijms, 2023, doi:10.3390/ijms241612630_

Round 1

Reviewer 1 Report

the manuscript is interesting, generally well written and well illustrasted. To my opinion, it can be accepted in the present form.

Reviewer 2 Report

The manuscript demonstrated Combined administration of escitalopram oxalate and  nivolumab exhibits synergistic growth-inhibitory effects on  liver cancer cells through inducing apoptosis The study is interesting; however, some of the results presentations are not clear; without them it would be prudent to publish the manuscript in present form.

1.    If the time-dependent IC50 doses for escitablopram oxalate and nivolumab alone are included by the authors, It would be useful.

2.    The authors kindly include the percentage of apoptosis and SubG1 for combination drug-treated cells in the results section.

3.    It would be excellent if the authors of this manuscript exhibited anti-apoptotic markers.

4.    Authors have to specify the magnification used to obtain the IHC pictures in figure 5 legends.

5.    Tunel pictures did not correspond to the outcome. The brown signal in the control photographs is higher than in the drug-treated photos. Explain.

Reviewer 3 Report

Heoretical basis for the combination index (CI)-isobologram equation that allows quantitative determination of drug interactions, where CI<1, =1, and >1 indicates synergism, additive effect and antagonism, respectively.

On the basis of this the paper needs some modifications/additions:

Materials and Methods (4.3)

Enter what a value equal to 1 indicates. If greater than 1?

Results (2.2)

Highlight that there are CI>1 values

Emphasize that with some combinations of concentrations of the two drugs an antagonistic effect may be present.

Highlight this in the paper.To enhance the interest of the article and its critical evaluation of the data from the different experiments, it is suggested to comment and compare the data obtained from the different experiments with those obtained from the calculation using the CompuSyn software. Highlight both cases of synergy and those antagonistic

Round 2

Reviewer 2 Report

Authors addressed all the comments appropriately
